# Vector Competence of Mediterranean Mosquitoes for Rift Valley Fever Virus: A Meta-Analysis

**DOI:** 10.3390/pathogens11050503

**Published:** 2022-04-23

**Authors:** Alex Drouin, Véronique Chevalier, Benoit Durand, Thomas Balenghien

**Affiliations:** 1Epidemiology Unit, Laboratory for Animal Health, French Agency for Food, Environmental and Occupational Health & Safety (ANSES), University Paris-Est, 94701 Maisons-Alfort, France; alex.drouin@anses.fr (A.D.); benoit.durand@anses.fr (B.D.); 2ASTRE, University of Montpellier, CIRAD, INRAE, 34398 Montpellier, France; thomas.balenghien@cirad.fr; 3CIRAD, UMR ASTRE, Antananarivo 101, Madagascar; 4Epidemiology and Clinical Research Unit, Institut Pasteur de Madagascar, Antananarivo 101, Madagascar; 5CIRAD, UMR ASTRE, Rabat 10101, Morocco; 6IAV Hassan II, UR MIMC, Rabat 10101, Morocco

**Keywords:** Rift Valley fever virus, mosquito, competence, meta-analysis, Mediterranean Basin

## Abstract

Rift Valley fever (RVF) is a zoonotic disease caused by a virus mainly transmitted by *Aedes* and *Culex* mosquitoes. Infection leads to high abortion rates and considerable mortality in domestic livestock. The combination of viral circulation in Egypt and Libya and the existence of unregulated live animal trade routes through endemic areas raise concerns that the virus may spread to other Mediterranean countries, where there are mosquitoes potentially competent for RVF virus (RVFV) transmission. The competence of vectors for a given pathogen can be assessed through laboratory experiments, but results may vary greatly with the study design. This research aims to quantify the competence of five major potential RVFV vectors in the Mediterranean Basin, namely *Aedes detritus*, *Ae. caspius*, *Ae. vexans*, *Culex pipiens* and *Cx. theileri*, through a systematic literature review and meta-analysis. We first computed the infection rate, the dissemination rate among infected mosquitoes, the overall dissemination rate, the transmission rate among mosquitoes with a disseminated infection and the overall transmission rate for these five mosquito species. We next assessed the influence of laboratory study designs on the variability of these five parameters. According to experimental results and our analysis, *Aedes caspius* may be the most competent vector among the five species considered.

## 1. Introduction

Rift Valley fever (RVF) is an arbovirosis caused by a *Phlebovirus* of the *Phenuiviridae* family (*Bunyavirales* order) [1]. RVF virus (RVFV) affects domestic ruminants such as cattle, sheep, goats or camels, and has major consequences in terms of health and economics. It is also able to infect wild animals [2]. In livestock, the clinical form of RVF depends on the age and physiological status of the animal. It is often asymptomatic in adults, but may cause a high abortion rate in pregnant females [3]. Mortality in young animals is high: between 10 and 70% for calves and up to 100% for lambs [4]. The disease also affects humans, giving rise to asymptomatic or non-specific flu-like symptoms in most cases, but complications may occur with ocular, neurologic and hemorrhagic symptoms [5,6]. RVFV is transmitted among ruminants by mosquitoes, mainly from the *Aedes* and *Culex* genera. However, direct transmission between animals can probably occur [7,8]. Humans are mostly infected through direct contact with infected livestock or with their tissues and fluids, but infections from mosquito bites are possible [7,9].

RVF was first described in Kenya in 1930 [10], and has been reported in southern Africa since the 1950s. The virus was then detected in Egypt in 1977 and in western Africa since the 1980s—outbreaks are regularly reported in Senegal and Mauritania, but concern the whole Sahel region. The first RVF outbreak reported outside Africa occurred in the Arabian Peninsula in 2000 [11]. RVFV was probably introduced into Egypt in 1977 and in 2003, and into Yemen in 2000 through ruminant trading [11]. Even though the livestock trade is strictly regulated between RVF endemic regions and most of the Mediterranean countries, uncontrolled live animal trade routes exist from sub-Saharan to North African countries [12,13]. It has been suggested that the movements of camels across the Sahara Desert could carry RVFV to northern Africa [14]. Illegal animal movements have been incriminated in the emergence of RVF in Libya, in southern Cyrenaica in 2019 [15]; the disease was then reported in the Fezzan region in 2020 and on the Mediterranean coast of Tripolitania in 2021 [16]. Moreover, serological evidence of RVF in ruminants has been found in western regions of the Sahara [17] and in Tunisia [18,19,20], where competent mosquito vectors are present [21,22,23,24]. This may suggest past or current RVFV circulation in these regions, or indicate that animals may have been introduced there after being infected in endemic areas.

The vector competence of an arthropod for a given pathogen relies on its ability to get infected by the pathogen, to support its replication and to transmit it to a vertebrate host [25]. The competence of mosquitoes for arboviruses varies between species and between populations of the same species. It may be influenced by intrinsic factors such as the genetics of both the virus and the vector, the innate immune response or the microbiota of the mosquito [26,27,28,29,30], but also by extrinsic factors, such as temperature or the viremia of the host [28]. After taking an infectious blood meal from a viremic animal, biological barriers can limit the viral course of the infection into the body of the mosquito. In competent vectors, the virus first infects the mesenteron and then disseminates in the tissues of the arthropod, including the salivary glands. The virus can then be transmitted to another host by bite. However, the existence of a midgut infection barrier (MIB) can prevent or limit the infection of the epithelial cells of the mesenteron. In addition, the midgut escape barrier (MEB) affects the dissemination of the virus to other organs from these cells. Finally, the salivary gland infection and escape barriers (hereinafter referred to as ‘salivary gland barriers’, SBs) can prevent the transmission of the virus into the saliva of the mosquito [27,28,31,32]. Biological barriers rely on complex virus–mosquito interactions and involve several mechanisms such as physical barriers, molecular interactions between the virus and the mosquito cells, enzymatic reactions or the immune system [27]. In the context of laboratory experiments on vector competence, the infection rate (IR) represents the proportion of mosquitoes exposed to an infected blood meal in which the virus has at least crossed the MIB. The dissemination rate among infected mosquitoes (DR/I) and the overall dissemination rate (DR) are the proportions of mosquitoes in which the virus has passed through the midgut and crossed the MEB, spreading through different tissues, such as in the wings, legs or head. The values are calculated among infected mosquitoes only or among all exposed mosquitoes, respectively. The transmission rate among mosquitoes having a disseminated infection (TR/D) represents the proportion of mosquitoes transmitting the virus, i.e., the proportion of mosquitoes in which the virus has crossed the SBs out of all the mosquitoes with a disseminated infection. The overall transmission rate (TR) represents the proportion of mosquitoes in which the virus has crossed all biological barriers, and thus quantifies the overall mosquito competence.

Laboratory experiments are of prime importance as part of the process of demonstrating the vector status of a given species for a given pathogen. In addition, a quantitative estimation of vector competence means that this parameter may be included in the calculation of vector capacity, defined as the ability of a vector population to transmit a pathogen in a given spatiotemporal context [25,28]. Nevertheless, the experimental design of laboratory experiments may affect the estimation of vector competence. It has been shown that estimated competence for RVFV may vary at least with the mosquito species, the viral strain, the viral titer of the infectious blood meal, the rearing temperature or the feeding method [33,34,35], leading to limitations when using estimates computed in a given context.

Meta-analysis is a statistical synthesis used to summarize data obtained from different studies by computing a weighted average of their results. Meta-regressions, which are part of the meta-analysis process, may also be used to quantify variability among these results and attempt to explain it [36,37,38]. This methodology is thus very useful in a context of multiplicity and variability of results, such as with experimental vector infections.

The aim of this study was to quantify the vector competence of the five main potential mosquito vectors of RVFV in the Mediterranean Basin using a meta-analysis methodology applied to published laboratory data, and to assess the variability of the five components of this competence, i.e., IR, DR/I, DR, TR/D and TR.

## 2. Materials and Methods

RVFV may be able to infect more than 50 species of mosquitoes, an assumption based on the presence (isolation or genome detection) of the virus in field-collected individuals [2,32]. The ability to transmit the virus by bite after oral exposure or intrathoracic inoculation has been demonstrated in the laboratory for at least 47 species [32], of which more than 10 are present in the Mediterranean Basin. Based on current entomological knowledge and data on observed/predicted presences [21,22,23,36,37,38,39,40,41], we selected *Aedes caspius*, *Ae. detritus sensu lato* (pooling together the sibling species *Ae. coluzzii* and *Ae. detritus sensu stricto*, hereinafter named ‘*Ae. detritus*’), *Ae. vexans*, *Culex pipiens* (pooling together the forms *Cx. pipiens pipiens* and *Cx. pipiens molestus*, hereinafter named ‘*Cx. pipiens*’) and *Cx. theileri* as the five mosquito species most likely to be vectors of RVFV should it spread across the Mediterranean Basin.

### 2.1. Article Search Strategy and Selection Process

To select articles of interest, a systematic review of the scientific literature was conducted according to the Preferred Reporting Items for Systematic Reviews and Meta-Analyses (PRISMA) guidelines [42,43]. Searches were performed in Scopus and PubMed databases using the ‘title, abstract, and keywords’ or ‘title and abstract’ fields, respectively. The Boolean query used was: (*mosquito* OR aedes OR culex OR caspius OR detritus OR coluzzii OR vexans OR pipiens OR theileri*) *AND* (*competen* OR disseminat* OR transmi**) *AND* (*rift AND valley AND fever*).

After removing duplicates, all the articles were considered, without any date or language restriction. The inclusion was performed by the same author in three steps: title screening, abstract screening and full-text reading. In the first two steps, articles were included only when they concerned at least one part of mosquito competence (infection, dissemination or transmission) tested in a laboratory context, and at least one of the five selected mosquito species. In the last step, exclusion criteria were: review articles, no competence measure extractable, data already used in another article or study design leading to results not comparable with other articles, even for the control group.

### 2.2. Data Extraction

Data concerning the five following parameters, subsequently named ‘outcomes’, were extracted from the selected articles: IR, DR/I, DR, TR/D and TR (Table 1). Even if these outcomes are not actual rates but proportions (as there is no time dependency), we chose to keep the commonly used term ‘rate’. Raw numerators and denominators of the five outcomes were extracted and used directly when they were available, or computed from the given percentages and size of the mosquito batches (i.e., groups of mosquitoes subjected to the same experimental conditions). When raw data were not available in the text or in tables, we estimated the proportions from bar charts provided in the articles using Webplotdigitizer 4.4, an online software program used to measure distances on images [44]. Outcomes that were not directly estimated by the authors in the article were calculated if the corresponding numerators and denominators were available. From the same article, data related to some batches may have been included, whereas other data were excluded because the associated experimental design was too specific to be compared with the other studies that had been included. Transmission rates that were provided for ‘infected’ mosquitoes without any information on dissemination were excluded from the pool of TR/D data. Mosquitoes that had been inoculated intrathoracically with RVFV were considered to have a disseminated infection, and were therefore included for the TR/D outcome.

Information on the experimental design was also collected, i.e., the mosquito species, viral titer of the blood meal, rearing generation of the mosquitoes, rearing temperature, viral strain used, number of days between exposure and outcome assays and the country of origin of the mosquito strain (Table 2). We did not consider the methods used for viral detection (immunological assays, plaque assays on Vero cells, reverse transcription polymerase chain reactions (RT-PCR), histological methods, or inoculation in mice). Data were transformed into categorical variables when relevant. For the viral titer of an infectious blood meal, data given in plaque forming units per milliliter (PFU/mL) were log-transformed and classified into a ‘low’, ‘low to medium’, ‘medium’, ‘medium to high’ or ‘high’ viral dose, using thresholds suggested in Lumley et al., 2018 [45]. If information was only given about the virus titrated in mosquitoes after a blood meal, we computed the viral titer of exposure assuming that a mosquito ingests 0.003 mL of blood (titer of blood meal = log(titer ingested/0.003)), as suggested by Turell and Rossi [46]. Data measured in 50% tissue culture infective dose per milliliter (TCID50/mL) were converted using the formula PFU/mL = TCID50/mL × 0.69, as in Golnar et al. [47]. As there is no universal conversion between (suckling) mouse intracerebral 50% lethal dose ((S) MICLD50/mL) and PFU/mL, data using this unit were classified into a ‘high’, ‘medium’ and ‘low’ viral dose according to expert opinion. For intrathoracically inoculated mosquitoes, the infectious dose was not taken into account. The rearing temperature was classified using 20 °C and 25 °C as thresholds. When the rearing temperature was provided as a range, a new corresponding class was created. Concerning the time period between the exposure and assays, we transformed all durations provided in a number of days post-exposure into weeks, and a new class was created if the information was given as a range (Table 2).

### 2.3. Statistical Analyses

The primary goal of our statistical analyses was to compute for each species a summary value (called a ‘summary effect size’) for each of the five outcomes (IR, DR/I, DR, TR/D and TR), and secondarily to assess the effect on this value of parameters describing the experimental design, also called ‘moderators’. For each outcome, the analysis was performed in three steps: (i) model selection, with the models using random effects and the outcome values as response variables; (ii) subgroup analyses to assess the species effect, then (iii) meta-regressions. All these analyses were implemented in R 4.0.3 [48] using restricted maximum likelihood (REML) estimation.

(i) Model selection: as there could be several effect sizes originating from the same article, we considered three nested levels of variability (also called ‘heterogeneity’): the sampling error, the within-study variability and the between-study variability. Based on likelihood ratio tests (LRT), we tested the significance of the within-study variability by comparing models having the outcome value as the response variable, no fixed effect, and either two levels of random effects (i.e., sampling error and between-study heterogeneity) or three levels of random effects (i.e., sampling error, between-study and within-study heterogeneity) [49]. The effect sizes were transformed to obtain a normal distribution using the Freeman–Tukey double-arcsine transformation, as several outcome values were equal to 0 or 100% [50,51].

The presence of significant residual heterogeneity was then assessed using the Q-test and I² statistic (the percentage of heterogeneity among the total variance). If there was no residual heterogeneity after accounting for random effects, we used the model based on random effects alone to calculate the summary effect size, as there was no statistical evidence of an effect of the mosquito species on the outcome. Otherwise, we proceeded to step (ii).

(ii) Subgroup analysis: we studied the influence of the mosquito species on the outcome by including the species as a fixed effect in the statistical model selected in step (i), and we assessed its effect using a test of moderators (omnibus test of coefficients [52]). In case of a significant effect, we next used the species-specific model to compute the summary effect size of that outcome for each species of mosquito and the corresponding 95% confidence intervals. Otherwise, we computed a single summary value for all species. Proportions were obtained from model coefficients using the inverse of the Freeman–Tukey double-arcsine transformation and the harmonic mean of the sample sizes [53].

Once random effects (step (i)) and possibly the species effect (step (ii)) had been accounted for, and if there was still residual heterogeneity, we proceeded to step (iii).

(iii) Meta-regression: we analyzed the influence of the study designs on the response variable by including the corresponding variables (called ‘moderators’) as fixed effects in the meta-analytic model. The moderators were the viral titer of the blood meal, viral strain, temperature of rearing, number of days post-exposure, generation of colonization of mosquitoes and the country of origin of the mosquito strain (Table 2). The moderators were first evaluated sequentially [51,54] by adding a variable as a fixed effect in the model resulting from step (i) or (ii). As no classical model selection methods are available for comparing models with different fixed effects estimated by REML, significant variables were selected through tests of moderators on the variable coefficients. All significant moderating variables were then added into one final multivariate model.

As a last step, we assessed whether the resulting model fully captured the heterogeneity of each outcome, or whether residual heterogeneity still existed.

## 3. Results

### 3.1. Article Inclusion and Data Extraction

Our initial query returned 953 results. After removing duplicates, 601 articles were considered at the title screening level, then 76 at the abstract screening level. Finally, 43 articles were fully read, and 34 were included in the meta-analysis (Figure 1). Nine articles were excluded during full-text reading. Two were review articles [47,55]. No competence measure was extractable from four articles (no quantitative results [56,57], or data were only available for pooled mosquitoes [58,59]). For two articles, the reported data were already used in another included study [60,61]. For one article, the study design led to results not comparable with other papers (co-infection experiments [62]). Only the control batch was included in the study for two articles (other batches concerned interrupted meals or immunized hosts [63], and RVFV coinfection with Flaviviruses [64]). Data extraction provided 182 batches of mosquitoes tested for IR, ranging from 0 to 100%, 95 for DR/I (0 to 100%), 171 for DR (0 to 90%), 65 for TR/D (0 to 100%) and 119 for TR (0 to 66.7%) (Table 3).

### 3.2. Statistical Analyses

#### 3.2.1. Model Selection

The three-level model, which takes into account sampling error, within-study and between-study heterogeneity, was the best model (LRT: *p* < 0.05) for all of the outcomes. There was still a considerable amount of residual heterogeneity as assessed by Q-tests (*p* < 0.05) and I^2^ (>75%) for all computed outcomes. A subgroup analysis was therefore performed to assess the influence of the mosquito species and to compute a summary effect size for each outcome.

#### 3.2.2. Subgroup Analysis: Influence of the Mosquito Species on Outcomes

We found an effect of the mosquito species on IR, DR/I, DR and TR/D (*p* < 0.05 for all tests of moderators in subgroup analyses). The effect size and 95% confidence intervals were estimated for these four outcomes and for each species if data were available (Table 4). *Aedes caspius* and *Ae. detritus* had a high IR, moderate DR/I and DR, and high TR/D with large confidence intervals for *Ae. detritus*. *Culex pipiens* and *Ae. vexans* had a moderate IR and low DR/I and DR. The TR/D was high for *Cx. pipiens* and moderate for *Ae. vexans*, with a large confidence interval. The IR of *Cx. theileri*, the only outcome computable for this species, was high with a large confidence interval. Finally, considering the absence of a significant effect of the mosquito species on TR, the overall computed summary value of this outcome was 9.8% [7.1; 12.9].

#### 3.2.3. Meta-Regression: Influence of Study Designs on Outcome Values

A significant amount of residual heterogeneity was present for all outcomes after taking into account the mosquito species effect, and meta-regressions have thus been conducted. Results of the final meta-regression models are provided in Table 5. The viral titer of the blood meal was a significant moderator for IR, with higher titers leading to higher rates of infection: the ‘low’, ‘low to medium’ and ‘medium’ classes had a significantly lower effect size than the ‘high’ class. The same effect was observed for DR, with the ‘medium’ class having a lower rate than the ‘high’ class. The rearing temperature was a significant moderator for IR, with higher temperatures leading to a higher IR: the ‘>25 °C’, ‘13 then 26 °C’ and ‘20 then 28 °C’ classes had a higher effect size than ‘<20 °C’. There was a significant difference between the countries of origin of the mosquitoes for TR/D. The rearing generation of mosquitoes was a significant moderator for DR/I and DR, with ‘>F5’ classes having significantly higher rates than ‘F0/F1’. We did not find any effect of the moderators on TR.

Finally, the Q-test was significant (*p* < 0.05) for models including all significant moderators, for all outcomes. These results suggest that a significant part of heterogeneity has not been explained by either the subgroup analysis or the meta-regressions.

## 4. Discussion

Very few articles have used the meta-analysis methodology to summarize data on experimental vector competence studies; they are related to Flaviviruses, namely dengue virus [90], Japanese encephalitis virus [91] or Zika virus [92]. Our study focused on five potential RVFV vector species in the Mediterranean Basin, namely *Ae. caspius*, *Ae. detritus*, *Ae. vexans*, *Cx. pipiens* and *Cx. theileri*, and summarized data from 34 laboratory studies on vector competence. For each vector, we computed summary values for the infection rate (IR), the dissemination rate among infected mosquitoes (DR/I), the overall dissemination rate (DR), the transmission rate among mosquitoes having a disseminated infection (TR/D) and the overall transmission rate (TR). Once infected by RVFV, the five mosquito species were able to transmit the virus to another host. *Ae. caspius* appeared to be the most competent vector, as it had the highest values for IR, DR and TR/D. Even though few data were available for *Cx. theileri*, published studies showed that 13 to 71% of the infected mosquitoes (without any information on their dissemination status) transmitted the virus to various mammals [88,89].

To our knowledge, only two studies have summarized data about vector competence of mosquitoes for RVFV. In Madagascar, Tantely et al. [55] focused on the 32 species present in the country and compiled data on IR and TR, but did not calculate a summary statistic of these values. Golnar et al. [47] used linear regression methods to compute the DR and TR/D of 26 mosquito species present in the USA, including only experiments conducted at 26 °C and for individuals exposed to 7.5 log PFU/mL of RVFV. Their results are consistent with ours. For *Cx. pipiens*, they obtained a DR of 13% and a TR/D of 90% (versus 13.5% [8.0; 20.0] and 93.6% [80.4; 100] in our study, respectively). For *Ae. vexans*, the estimated TR/D was 41% (versus 38.3% [14.4; 64.7] in our study). They computed a higher DR (26% versus 13.6% in our study), but this value was calculated from two publications rather than eight in our study.

Several biological barriers limit the viral spread in the body of the mosquito. Previous work hypothesized the existence of a midgut infection barrier (MIB) in *Cx. pipiens* because the value of IR was lower than 100% [78]. Similarly, the existence of mosquitoes with or without a disseminated infection has been pointed out as proof of the existence of a midgut escape barrier (MEB), preventing viral escape from the mesenteron in some individuals or perhaps only delaying it for up to a few weeks [80]. In this study, we found a moderate IR and relatively low DR/I and DR for *Cx. pipiens*, supporting the hypothesis that the MIB and MEB exist for this species, with the MEB playing a major role: only 22.3% of the *Cx. pipiens* mosquitoes with an infection of their mesenteron finally had a disseminated infection. Midgut barriers appear to be the most important determinants of vector competence in *Cx. pipiens* and other *Culex* species [66,79,80,84]. We furthermore calculated a high TR/D for *Cx. pipiens*, with 93.6% of the mosquitoes with a disseminated infection being able to transmit the virus, leading to the hypothesis of quasi-inexistent salivary gland barriers (SBs). This is supported by previous findings [24,66,79,80,84], except one experimental infection carried out with field-collected mosquitoes from Lebanon [83]. For *Ae. vexans*, the MIB and MEB have been demonstrated to be moderate to severe, and SBs were qualified as inexistent to moderate depending on the study [35,67,70]. This is consistent with our results, showing that this species is probably a moderately competent vector.

In addition to estimating vector competence, we tried to explain the variability of its estimation between published studies using meta-regressions. First, we highlighted that IR was dependent on the viral titer of the blood meal, as previously demonstrated [24,35,45,66,82]. Surprisingly, a higher viral titer led to higher values of DR but not of DR/I. In fact, this relation between the viral dose and dissemination has been demonstrated experimentally [24,35,45,66] and linked to the existence of a dose-dependent midgut escape barrier in some species [45]. As IR and DR are dose-dependent, it was expected that TR (measuring the overall cycle of infection, dissemination and transmission of the virus) would increase with the viral titer of the blood meal, as observed for some species in [45,66,93]. However, there was no significant effect of any of the moderators tested in our analysis on TR (mosquito species, viral titer of the blood meal, rearing generation of mosquitoes, rearing temperature, viral strain used, number of days between exposure and outcome assays and the country of origin of the mosquito strain). This could be explained either by a lack of statistical power and/or to different, opposite and species-specific barrier effects. The absence of effects of any moderator on TR was also observed by Oliveira et al. for Japanese encephalitis virus [91].

We did not find any effect on infection, dissemination or transmission of the period between the exposure of the mosquitoes to the virus and experimental assays. These results are again surprising because dissemination has been demonstrated to be time-dependent [93], with the midgut barrier delaying viral dissemination into the body of the mosquito [84]. Moreover, the virus can pass through the midgut barriers at various times after infection [74,84,85]: a mosquito that does not have a disseminated infection at a given point in time after infection may develop it later [74]. Our results may be due to the poor precision of data concerning the time between exposure and assays that are often provided as a range rather than a time point, especially for transmission experiments (TR/D and TR). More precise reporting of this delay in further competence studies would be beneficial in assessing the effect of the time parameter.

As mosquitoes are ectotherm organisms, temperature is one of the most important abiotic factors influencing both their biology and virus transmission (reviewed in [26,29]). It notably affects the extrinsic incubation period (EIP), i.e., the time between the infection of a mosquito and its ability to transmit the pathogen: a higher temperature is associated with a shorter EIP and increased transmissibility of the virus [26,29]. Nevertheless, temperature also affects the viral replication and immune response of the mosquito, and the relationship between competence and this parameter is complex: higher rates of infection or transmission have been demonstrated for lower temperatures, with increased mosquito mortality and enhanced barriers at higher temperatures [26,29,32]. In our study, this variable had a significant effect on IR, with a higher temperature leading to higher rates, but not on either DR or DR/I. This is consistent with the conclusions of Turell et al. in their study on *Cx. pipiens* [34]: EIP was inversely related to the rearing temperature, with higher values leading to an increase in IR, but only leading to a decrease in the duration of dissemination without affecting the rate (DR/I). However, the results for *Cx. pipiens* cannot be extrapolated to other species. In the same study, the authors did not find any effect of temperature on IR in *Ae. taeniorhynchus* but found a positive one on DR/I, as further confirmed in [94]. In our study, we found no significant effect of temperature on either TR/D or TR. Again, this may be linked to a lack of statistical power due to the multiplicity of the variable classes.

Some of the variability in vector competence among the studies included in our meta-analysis could be explained by the diversity of origins of the mosquito strains. We combined data obtained from mosquitoes originating from all geographic regions and highlighted significant differences in the TR/D between mosquitoes from Egypt and those from Lebanon and the UK. This is consistent with results for *Ae. vexans*, which was considered a moderate experimental vector of RVFV in the southeastern part of the USA and in Senegal, but almost incompetent in the northern and western part of the USA and in Canada [35,68,69,71,72]. As the vector competence of mosquitoes is at least partly under genetic control, the disparities observed are probably related to the genetic diversity of mosquito populations and to their specific relations to viral strains [28,95]. In particular, these variations may concern the efficiency of the MIB and MEB [71]. Geographic differences in vector competence have also been shown for other viruses of the *Bunyavirales* order, such as the La Crosse virus [31], or of the *Flaviviridae* family, such as dengue virus [28].

The effect of laboratory colonization on vector competence for RVFV has been studied by Gargan et al. [76]. The authors showed a correlation between colonization and IR, with the rate increasing with the number of rearing generations. However, they showed the opposite correlation with the percentage of infected mosquitoes (regardless of dissemination status) transmitting the virus. These results are supplemented by those of Moutailler et al. [22], who observed a higher DR in long-term established colonies compared to field-collected mosquitoes, and by unpublished data describing a decrease in TR among mosquitoes reared in laboratory conditions for several generations, reported in [77]. An increase in IR with laboratory colonization has been observed in *Aedes albopictus* and *Aedes aegypti* for dengue virus [96], whereas a decrease has been reported in *Ae. aegypti* for Yellow Fever virus [97]. The effect of colonization is probably linked to genetic selection or to mosquito microbiome changes [98]. In our study, we observed a significant effect of laboratory colonization on the DR/I and DR, as observed by Moutailler et al. [22], but not on any other outcomes.

Mosquito competence may vary with the viral genotype [28]. In this work, we included studies using attenuated strains of RVFV [22,46,73,82,83,84] even though a lower dissemination rate has been evidenced for Clone 13 [22]. However, we took this potential effect into account by including a specific viral strain moderator in the meta-regressions. Our results showed no differences between this strain and ZH501, which was the most used one. No significant differences between ZH501 and any other strains were found either. In the literature, authors have not found any significant difference in infection, dissemination or transmission between the strains ZH501 and Lunyo [45], but transmission only occurs for SH172805 and not for either AnD 133719 or ArD 141967 [69].

Diversity in the experimental designs, in the viral strains used and in the origin of the mosquitoes tested all lead to great variability in parameter estimations, making the experimental results difficult to extrapolate. Our study explained only part of the heterogeneity observed in the data, and there are potential sources of heterogeneity other than those integrated in this analysis, such as feeding methods (live animals or artificial feeding systems) [33,99], methods used to measure the transmission rate (feeding on hamsters, PCR with saliva) or viral assay techniques (immunofluorescence, complement fixation reaction, plaque assay on Vero cells, PCR). The quantification of vector competence is crucial to gain insight into the ability of mosquitoes to transmit RVFV and their role in the circulation of the virus. It is also needed to parametrize mathematical models of transmission that will be further used to perform risk assessment, unravel epidemiological mechanisms or anticipate the consequences of the virus being introduced into a disease-free region. A standardization of experimental designs is therefore important to improve the comparability of results and to better estimate the parameters. Besides, more data are needed to estimate the five studied outcomes with a higher precision, especially for DR, TR/D and TR, which are of major importance to assess the effect of biological barriers and the whole competence of the mosquito species. Moreover, we highlighted a lack of data concerning *Cx*. *theileri*. This species, which appeared to be an efficient RVFV vector in South Africa [88,89], is present throughout the Mediterranean Basin, and is particularly abundant in northern Africa. Further studies should investigate its potential for transmission.

As a whole, our analysis confirms that the five species of concern could be involved in RVFV transmission in the Mediterranean Basin, and should prompt both further investigation into the risk of RVFV being introduced into what was previously an RVFV-free region, and reinforced surveillance in high-risk areas.

## Figures and Tables

**Figure 1 pathogens-11-00503-f001:**
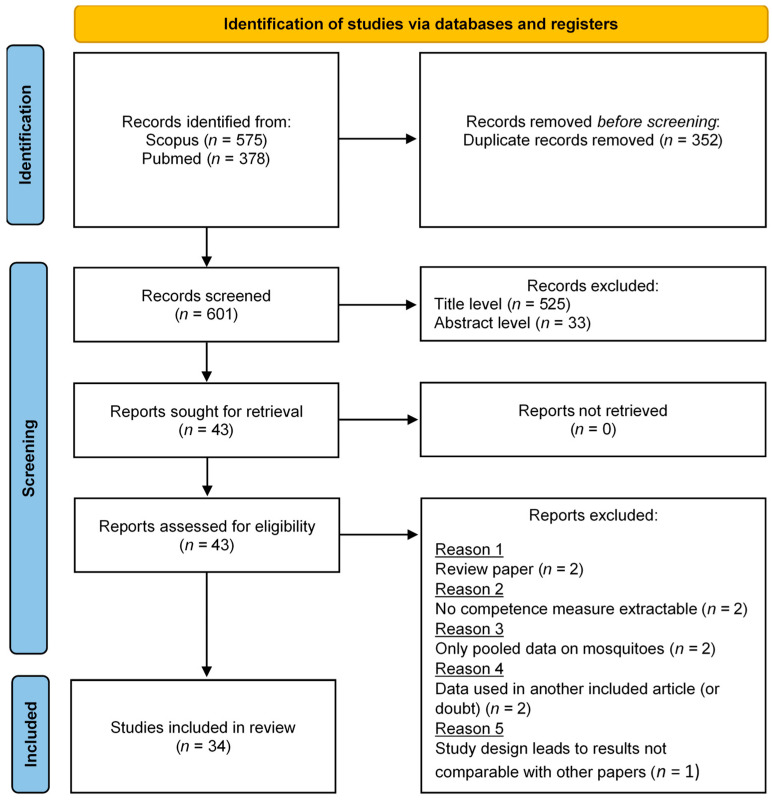
PRISMA flow diagram representing the article selection process (from [42]).

**Table 1 pathogens-11-00503-t001:** Outcomes of interest studied in the meta-analyses.

Outcome	Definitions	
	Numerator	Denominator
Infection rate (IR)	Number of infected mosquitoes (virus detected in the whole mosquito, or in the body without including legs and wings)	Number of mosquitoes fed
Dissemination rate among infected mosquitoes (DR/I)	Number of mosquitoes with a disseminated infection (virus detected in legs, wings or head squashes)	Number of infected mosquitoes
Overall dissemination rate (DR)	Number of mosquitoes with a disseminated infection (virus detected in legs, wings or head squashes)	Number of mosquitoes fed
Transmission rate among mosquitoes having a disseminated infection (TR/D)	Number of mosquitoes that transmit the virus to another host (or contain virus in saliva or salivary glands)	Number of mosquitoes with a disseminated infection or Number of mosquitoes that had been inoculated intrathoracically with RVFV
Overall transmission rate (TR)	Number of mosquitoes that transmit the virus (or contain the virus in saliva or salivary glands)	Number of mosquitoes fed

**Table 2 pathogens-11-00503-t002:** Moderating variables studied in the meta-analyses.

Moderating Variable	Definition	Classes *	
Mosquito species	Taxon to which the tested individuals belong	*Aedes caspius* *Aedes detritus* *Aedes vexans*	*Culex pipiens* *Culex theileri*
Viral titer of the blood meal	Titer of virus in the blood on which the mosquitoes took their blood meal, i.e., host viremia in the case of a live host or titer of virus in the artificial feeder	Low Low to medium Medium	Medium to high High
Rearing generation of mosquitoes	Field-collected or colonized lines of mosquitoes	F0/F1 (field-collected mosquitoes or first generation) F2/F5 (colonized mosquitoes between second and fifth generation) >F5 (mosquitoes over five generations of laboratory colonization)	
Rearing temperature	Temperature at which mosquitoes were kept during incubation	<20 °C 20–25 °C >25 °C 13 then 26 °C	20 then 28 °C 22–26 °C 26 °C (day)/22 °C (night)
Viral strain	Viral strain used to infect mosquitoes	AN 1830 AnD133719 ArD141967 Clone 13 Kenya-128B-15 Lunyo RVF MP-12 SH172805	strain 35/74 strain 56/74 T1 unknown (‘wild type’) ZH501 ZH501 or Dak ArB 1976 ^†^ ZH501 or Egypt93 ^†^ ZH548
Time period between exposure and assays	Time between infectious blood meal and assay for mosquito infection (or dissemination or transmission)	*Period known precisely* ≤1 week >1 to ≤2 weeks >2 to ≤3 weeks >3 to ≤4 weeks >4 weeks	*Period provided as a range* <1 to ≤2 weeks <1 to ≤3 weeks <1 to ≤4 weeks >1 to ≤3 weeks >1 to ≤4 weeks >1 to >4 weeks >2 to 4 weeks
Country	Country of origin of the mosquito strain	Algeria Canada Cyprus Egypt UK France Germany Lebanon	Morocco Netherlands Senegal South Africa Spain Tunisia USA

* See text for class definitions. ^†^ Both viral strains are used in the same study without distinction.

**Table 3 pathogens-11-00503-t003:** Number of mosquitoes (*n*) and batches (*N*) included in the study for each outcome and each species.

Species	IR	DR/I	DR	TR/D	TR
** *Ae. caspius* **	*n* = 130 (*N* = 6)	*n* = 31 (*N* = 4)	*n* = 102 (*N* = 5)	*n* = 8 (*N* = 3)	*n* = 11 (*N* = 2)
	[45,65,66]	[45,66]	[22,45,66]	[45,66]	[45,66]
** *Ae. detritus* **	*n* = 118 (*N* = 8)	*n* = 25 (*N* = 4)	*n* = 121 (*N* = 10)	*n* = 2 (*N* = 2)	*n* = 112 (*N* = 8)
	[45]	[45]	[22,45]	[45]	[45]
** *Ae. vexans* **	*n* = 911 (*N* = 29)	*n* = 436 (*N* = 25)	*n* = 1843 (*N* = 31)	*n* = 129 (*N* = 19)	*n* = 655 (*N* = 20)
	[35,67,68,69,70,71,72]	[35,67,68,69,70,71,72]	[22,35,67,68,69,70,71,72,73]	[35,67,69,70,71,72]	[67,69,70,71,72]
** *Cx. pipiens* **	*n* = 6221 (*N* = 131)	*n* = 1593 (*N* = 62)	*n* = 4832 (*N* = 125)	*n* = 497 (*N* = 41)	*n* = 2453 (*N* = 89)
	[24,33,34,45,46,63,64,66,68,71,74,75,76,77,78,79,80,81,82,83]	[24,34,45,46,63,64,66,68,71,74,78,79,81,83]	[22,24,34,45,46,63,64,66,68,71,74,78,79,80,81,83,84,85]	[24,34,45,46,64,66,68,71,74,79,83,85,86]	[24,45,46,64,66,68,71,74,75,77,80,82,83,84,85,87]
** *Cx. theileri* **	*n* = 359 (*N* = 8)	*n* = 0 (*N* = 0)	*n* = 0 (*N* = 0)	*n* = 0 (*N* = 0)	*n* = 0 (*N* = 0)
	[88,89]				

Details on data extracted for each outcome and each mosquito species, and a list of references included, are given in Appendix A.

**Table 4 pathogens-11-00503-t004:** Summary values of IR, DR/I, DR, TR/D and TR for each mosquito species using subgroup analysis.

Species	IR (%)	DR/I (%)	DR (%)	TR/D (%)	TR (%)
*Ae. caspius*	96.7 [77.9; 100]	53.7 [20.5; 85.4]	34.3 [15.3; 56.1]	96.1 [50.9; 100]	9.8 [7.1; 12.9]
*Ae. detritus*	82.4 [61.2; 97.0]	65.4 [29.3; 94.6]	33.0 [17.4; 50.5]	78.3 [4.5; 100]	
*Ae. vexans*	40.7 [22.4; 60.2]	24.1 [9.4; 42.0]	13.6 [6.0; 23.2]	38.3 [14.4; 64.7]	
*Cx. pipiens*	68.0 [56.7; 78.3]	22.2 [12.5; 33.3]	13.5 [8.0; 20.0]	93.6 [80.4; 100]	
*Cx. theileri*	88.6 [56.9; 100]	NA	NA	NA	NA

95% confidence intervals are given in square brackets. An overall summary value has been computed for TR as there was no significant effect of the mosquito species on this outcome.

**Table 5 pathogens-11-00503-t005:** Effect of study design moderators on the five outcomes evaluated by the final meta-regression models.

Outcome	Moderator	Moderator Class	Coefficient	IC 95% (Lower Bound)	IC 95% (Upper Bound)	*p*-Value	
**IR**	Species	*Ae. caspius*	Reference ^†^				
		*Ae. detritus*	−0.216	−0.488	0.056	0.12	
		*Ae. vexans*	−0.635	−1.014	−0.256	0.001	*
		*Cx. pipiens*	−0.382	−0.609	−0.154	0.001	*
		*Cx. theileri*	−0.164	−0.683	0.355	0.536	
	Viral titer of blood meal	High	Reference				
		Low	−0.711	−0.96	−0.462	<0.001	*
		Low to medium	−0.698	−1.102	−0.295	0.001	*
		Medium	−0.272	−0.355	−0.189	<0.001	*
		Medium to high	−0.053	−0.571	0.465	0.84	
	Rearing temperature	<20 °C	Reference				
		>25 °C	0.467	0.191	0.743	0.001	*
		13 then 26 °C	0.587	0.267	0.908	<0.001	*
		20–25 °C	0.039	−0.484	0.562	0.884	
		20 then 28 °C	0.644	0.158	1.129	0.009	*
		22–26 °C	0.068	−0.56	0.696	0.832	
		26 °C (day)/22 °C (night)	0.097	−0.357	0.552	0.675	
	Viral strain	ZH501	Reference				
		AN 1830	−0.001	−0.634	0.633	0.998	
		AnD133719	0.1	−0.458	0.658	0.725	
		ArD141967	0.063	−0.499	0.625	0.826	
		Clone 13	−0.295	−0.755	0.164	0.207	
		Kenya-128B-15	0.146	−0.431	0.722	0.62	
		Lunyo	−0.058	−0.218	0.103	0.481	
		RVF MP-12	−0.047	−0.439	0.345	0.814	
		SH172805	0.457	−0.102	1.016	0.109	
		Strain 35/74	−0.253	−0.813	0.308	0.377	
		T1	−0.187	−0.585	0.211	0.358	
		ZH501 or Egypt93	−0.061	−0.524	0.402	0.797	
	Country	Egypt	Reference				
		Canada	−0.373	−0.974	0.228	0.223	
		Lebanon	0.015	−0.612	0.641	0.964	
		USA	−0.045	−0.348	0.258	0.772	
**DR/I**	Species	*Ae. caspius*	Reference				
		*Ae. detritus*	0.238	−0.191	0.667	0.278	
		*Ae. vexans*	−0.176	−0.523	0.171	0.32	
		*Cx. pipiens*	−0.401	−0.719	−0.083	0.014	*
	Rearing generation of mosquitoes	>F5	Reference				
		F0/F1	−0.259	−0.504	−0.013	0.039	*
**DR**	Species	*Ae. caspius*	Reference				
		*Ae. detritus*	0.059	−0.175	0.292	0.623	
		*Ae. vexans*	−0.239	−0.45	−0.028	0.027	*
		*Cx. pipiens*	−0.296	−0.485	−0.108	0.002	*
	Rearing generation of mosquitoes	>F5	Reference				
		F0/F1	−0.167	−0.266	−0.069	0.001	*
		F2/F5	−0.148	−0.373	0.078	0.199	
	Viral titer of blood meal	High	Reference				
		Low to medium	−0.37	−0.749	0.01	0.057	
		Medium	−0.186	−0.274	−0.099	<0.001	*
		Medium to high	0.106	−0.257	0.468	0.568	
**TR/D**	Species	*Ae. caspius*	Reference				
		*Ae. detritus*	0.006	−0.73	0.742	0.987	
		*Ae. vexans*	−0.364	−0.975	0.247	0.243	
		*Cx. pipiens*	0.027	−0.343	0.397	0.885	
	Rearing temperature	<20 °C	Reference				
		>25 °C	0.122	−0.113	0.356	0.309	
		13 then 26 °C	0.283	−0.015	0.58	0.063	
		20–25 °C	0.215	−0.24	0.67	0.354	
		22–26 °C	0.338	−0.25	0.925	0.26	
		26 °C (day)/22 °C (night)	−0.259	−0.661	0.142	0.205	
	Country	Egypt	Reference				
		UK	−0.579	−1.108	−0.051	0.032	*
		Lebanon	−0.638	−1.226	−0.05	0.034	*
		Senegal	−0.509	−1.042	0.025	0.062	
		USA	−0.243	−0.697	0.211	0.294	

Redundant moderator classes were dropped from the models. ^†^ ‘Reference’ refers to the class with which the others were compared in the model. * *p*-value < 0.05.

## Data Availability

The data used for this meta-analysis were collected from published studies and are available in Appendix A.

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
