# Peer review of "Vector Competence of Mediterranean Mosquitoes for Rift Valley Fever Virus: A Meta-Analysis"

_pathogens, 2022, doi:10.3390/pathogens11050503_

Round 1

Reviewer 1 Report

Overall, this was a scientifically sound study of high significance and a well written manuscript.

A few minor edits for the authors:

1) Correct reference errors in lines 136,157, 177, 220, 243,

2) Fix bullet alignment in table 2 so that are bullets within each cell are aligned with each other

3) Add the word "be" to line 260 so that text reads, "There will still be a considerable amount of residual heterogeneity"

4) In table 5, in row "Kenya-128B", change commas to decimal points in right three columns

5)in line 306, change "spread" to "spreads"

Reviewer 2 Report

The manuscript by Drouin et al. is a meta-analysis of Rift Valley fever virus vector competence studies. The authors’ inclusion criteria and definitions throughout the manuscript are well defined and consistently used throughout the manuscript, I like the use of Table 1 these terminology are often used interchangeably and incorrectly in the field. The article is well written, for the most part, and adds value to existing published experimental findings. In the absence of standardised methods for performing vector competence studies these analyses are important to the field. I have listed minor comments below for consideration to improve the manuscript

Introduction

Line 45: Change Southern to South (or remove capital on southern)

Line 63: Rephrase, this sounds a little odd “transmit it in return”

Line 67: Rephrase “blood meal’s”, does the blood meal possess an infectious dose? Perhaps use the host’s viraemia?

Line 69: “mosquito’s organism” do you mean organs? Used a few times in the text

Line 72: “The virus is next able”, as written it assumes a competent host but this is not stated. This also anthropomorphises the virus, viruses don’t have abilities. Perhaps change to in a competent host the virus next enters the salivary glands. Salivary gland infection also does not preclude transmission as there maybe salivary gland escape barriers.

Line 73: The text would benefit from a sentence or two describing some of the mechanisms of these barriers “after crossing the SB”, crossing is a bit of a simplification, is it physical, enzymatic or binding compatibility, time dependent etc. ?

Line 78: Spreading through different tissues”, sound vague we often use measurements of virus in the wings, legs, head as a proxy for dissemination in experiments.

Methods

Line 136: Check citation manager, “error! Reference source not found” there are a few further instances later in the text

Line 168: The conversion rate for PFU to TCID is x0.69 not 60.69 as stated

Line 169: This assumes a 1:1 ratio of artificial blood meal equivalence to feeding on live animals, the literature suggests underestimation of competence using blood meals e.g. Richards 2007 Reduced infection in mosquitoes exposed to blood meals containing previously frozen flaviviruses- there are more recent articles too, consider mentioning as a limitation/confounding variable.

Results

  • The tables would benefit from legends briefly describing the methods/ what the values mean in layman’s terms
  • Table 2: Some of the viruses are listed multiple times or using strain synonyms. In some cases multiple viruses are listed under one bullet point? This isn’t clear why, perhaps multiple strains were used in one article? but this is not consistent as some have been separated? Can the author’s clarify and if this is by design detail it in a legend?
  • Can the authors define what they mean by batches?
  • Table 4: A single value is given for TR? Not for each species is this correct? I can’t see an explanation for this but maybe I missed it in the text. Having an overall TR for each species would be a really useful statistic. I appreciate the results are quite descriptive as it is statistical analyses but I think again Table legends may help the reader. I understand these to be the summary values that are described in the text so include the term “summary values” in the table title to signpost it
  • Table 5: It seems the data are cleaned up here and the duplication in virus strains, previously seen in Table 2 are removed. I’m not sure why it reads “ZH501, Egypt 93” at the bottom of the first page of table though? Formatting issue maybe?

Discussion

The discussion as a whole is very descriptive, more in a style suited to results. The discussion would benefit from more critical analysis and linking the findings together, it currently reads as a long list.  Some thoughts for consideration- Are the results an artefact of study designs? do the authors have any recommendations for experimental design based on their findings? Can you suggest what would be significant numbers to recommend researchers to use? What is the bigger picture, does this affect how we control mosquitoes to control introduction of disease? What single factor in vector competence do you deem to be most important from your results?

Line 299; add related to Flaviviruses namely; … dengue also should not be capitalised.

Line 323: rephrase so there is no apostrophe for virus, viral spread perhaps?

Line 323-325 does not have a reference

What is the significance of the midgut barrier in the course of disease? Relate it to the bigger picture.

Line 329: change indicating to supporting

Aedes vexans- what is the overall effect of this combination of midgut and salivary gland barrier does it make it a good vector overall? Do future studies need to focus on one area more than another. These complete studies of IR, DR and TR are timely and resource intensive could more value be given by focussing on infection studies in some species vs intrathoracic injection to focus more on salivary gland barriers?

Lines 340-350: None of this is related to your own research, if there are insufficient numbers in the meta-analysis for these species to draw a conclusion, should they be included in the manuscript?

Lines 351-352: In the discussion this needs to be translated from mathematical terminologies for more accessibility to your findings.

Line 360: list the moderators used here that do not affect the TR, assume your readers have not read the rest of the manuscript.

Line 362: Arguably TR is the single most important value when we think about spread of arboviruses, this area needs expansion. What did other meta-analyses find? was TR influenced by factors- for example the Flavivirus studies that are in greater number than studies of RVFV will presumably have more power? Is your finding an artefact of limited data points?

Line 36: Were the n numbers sufficient at early time points to make this conclusion?

Line 403: dengue

Reviewer 3 Report

The study is very well written and interesting. 

Abstract

Line 14 - Arbovirosis is a word not commonly used. It may be easier for the reader to understand the word "disease" in this context.

Introduction

Line 43 - Infections occur through direct contact with not only infected livestock but their tissues or fluids (ie aborted fetuses or during slaughter).

Line 59 - "... or at least that susceptible animals..." This is unclear. Are you suggesting that the serological evidence is from animals brought into the area from endemic areas that were exposed prior to arrival or that an infected animal from an endemic area that was transported to the western Sahara regions enabled limited local transmission by local competent vectors? 

Line 69 - Do you mean the mosquito's organs?

Material and method

Line 136 and elsewhere - please ensure that the references display properly.

Table 1 - There is no mention in the text of Table 1.  Please make sure the columns are spaced sufficiently so that they do not overlap and check for appropriate breaks of words.

Table 2 - There is no mention in the text of Table 2. Perhaps this table would be more easily read in landscape. Please ensure adequate space between columns, check for appropriate breaks of words and ensure the alignment of the variables in the Classes category.

Results

Table 3 - There is no mention in the text of Table 3. The addition of the references makes the table hard to read, especially for Cx. pipiens. Perhaps note that they are available in the supplementary table and remove them from this table.

Line 260 - "...will still..." Was?

Table 5 - Please add either here or in the methods section what the label "Reference" means. It is unclear to the reader. It is hard to determine where each moderator begins and ends due to the lines. Perhaps only including a line after the last moderator class for each moderator would help. Please add headings to the second page of the table to help the reader. 

Discussion

Line 323, 366 - organs?

Beginning on line 333 and following the citations are in a different font than the rest of the paper. 

Line 403 - dengue is not capitalized

Line 420, 445, 447 - It should be RVFV not RVF on these lines because you are speaking of the virus not the disease
